# Influence of Silicone Additives on the Properties of Pressure-Sensitive Adhesives

**DOI:** 10.3390/ma15165713

**Published:** 2022-08-19

**Authors:** Karolina Mozelewska, Adrian Krzysztof Antosik

**Affiliations:** Department of Chemical Organic Technology and Polymeric Materials, Faculty of Chemical Technology and Engineering, West Pomeranian University of Technology in Szczecin, Piastow Ave. 42, 71-065 Szczecin, Poland

**Keywords:** pressure-sensitive adhesives, thermal crosslinking, silicone additives

## Abstract

Research was carried out on the influence of various silicone compounds on the properties of pressure-sensitive adhesives. Silicone-based pressure-sensitive adhesives have good self-adhesive properties and are used in many different industries. However, their thermal resistance is relatively low. In order to improve this property, modifications were made to these adhesives. Compositions were tested, such as viscosity or thermogravimetric analysis, as well as tests of finished products in the form of self-adhesive tapes, i.e., peel adhesion, tack, cohesion at room and elevated temperature, SAFT test (Shear Adhesive Failure Temperature), pot-live (viscosity) and shrinkage. During the tests, an increase in thermal resistance (225 °C), lower shrinkage (0.08%), and lower viscosity was achieved (16.5 Pas), which is a positive phenomenon in the technology of pressure-sensitive adhesives. Thanks to this research, the properties of silicone self-adhesive adhesives have been significantly improved.

## 1. Introduction

Pressure-sensitive adhesives are becoming more and more popular due to their very good properties, e.g., good resistance to aging, easy removal of adhesive from the surface, and good adhesion to polar substrates. They are defined as a special group of materials that exhibit significant adhesive forces and tack when in contact with the substrate, without the need for a chemical reaction. They are very important in everyday life. Scientists have been working for many years to replace certain joining methods (e.g., welding, riveting) using adhesives and adhesive tapes. As a result, there is continuous development in this industry area [1,2].

Pressure-sensitive adhesives applied as a thin layer on the carrier (paper, foil, fabric) have a very long life and practically do not age. They are applied to the rollers in liquid form and after evaporation of the solvent (solvent-based adhesives) or cooling (solvent-free adhesives) [3].

Today, pressure-sensitive adhesives are used in virtually all areas of life. In addition to the obvious uses, such as labels and office supplies, they also have many other interesting uses. They are used, for example, in construction as masking tapes and as protective films for the temporary protection of glass surfaces, building elements and car parts. For such applications, it is important to remove them without damaging the stuck surface. Pressure-sensitive adhesives in the form of double-sided tapes and foams are also used in microelectronics as alternative products to conventional adhesives. The labeling and outdoor applications market has been practically dominated by PSA. One cannot forget about the use of pressure-sensitive adhesives in the medical devices industry, in the form of one-sided and double-sided tapes for medical devices, bags for diagnostics or waste management. Bandages, skin-electrode attachments, wound closure materials, and surgical dressings are manufactured using adhesive tapes [4,5,6].

Due to their chemical composition, silicone adhesives differ significantly from other organic polymer adhesives. The main structure of the silicone adhesive consists of a polysiloxane skeleton composed of silicon-oxygen bonds. The energy of such a bond is much higher than the energy of the C-C bond [7]. For this reason, silicones exhibit unique properties in terms of resistance and stability. In addition, they are also characterized by high chain flexibility, due to the significant bond length and flexible bond angle [8]. The methyl groups attach the Si atoms, and they are steric undisturbed and can rotate freely, and as a result, the main polar chain -Si-O-Si- is shielded and allows only a few intra- and intermolecular interactions. That is why polysiloxane chains require chemical cross-linking with high cross-link density in order to obtain the best mechanical properties [9].

Silicone adhesives were introduced to the market in the 1960s and have been used in many different industries. Typical uses are: various industrial operations such as masking, wrapping, and bonding, but also in the electrical, electronics, healthcare, medical or automotive industries [10]. Since 2000, there has been a great deal of interest in new applications of silicone pressure-sensitive adhesives, especially medical and industrial tapes. Most traditional silicone pressure-sensitive adhesives are solvent based [11]. However, there is a growing need to reduce VOC (Volatile Organic Compound) emissions from tape and label manufacturers. Companies are looking to introduce new high solids pressure sensitive adhesives or hot melt types to reduce or eliminate VOC emissions. In addition, there is a trend towards more energy-efficient solutions on the market [12]. For pressure-sensitive adhesives, this is the increased development of addition-cure pressure-sensitive adhesives which can be cured at about 100 °C instead of 150 °C (required for typical cross-linkers) [13].

In the presented work, silicone adhesives were used, usually consisting of silicone polymers (with silane groups) and siloxane resins. They are specialist use materials, they show excellent performance at high and low temperatures, have excellent electrical properties, and have good chemical and weather resistance [14]. Silicone-based pressure-sensitive adhesives are characterized by high self-adhesive properties, although the thermal resistance is often relatively low. This is similar to pot-life, i.e., viscosity measured over time. Silicone adhesives without modification have a short pot life, they are only suitable for coating immediately after mixing. The shrinkage of silicone pressure-sensitive adhesives is also high and exceeds 1%, which is unacceptable in the technology of producing pressure-sensitive adhesive tapes and adhesives. The aim of the study was to determine the influence of various silicone additives on the properties of pressure-sensitive adhesives. The research was carried out to improve such properties as: thermal resistance, pot-life, and shrinkage.

In the presented work, the influence of various additives on the properties of silicone pressure-sensitive adhesives was investigated. They were: PDM OM 50, PDMS 500, and silicone oils: OS 300, SILOL-350F. The innovation of the article is the determination of the influence of various silicone additives: polydimethylsiloxanes polymers and various silicone oils. This influence has not yet been described in the literature available to the authors.

## 2. Materials and Methods

### 2.1. Materials

The research used: solvent-based silicone pressure-sensitive adhesive—DOWSIL™ 7358 (Q2-7358) from Dow Corning (USA), toluene from Carl Roth (Germany), and crosslinking agent—dichlorobenzoyl peroxide—DClBPO of Gelest (USA).

Used additives:PDMS OM 50—polydimethylsiloxane polymers with kinematic viscosities of 50 mm^2^/s,PDMS 500—linear polydimethylsiloxane polymers with kinematic viscosities of 500 mm^2^/s,OS 300—mixture of octamethylcyclotetrasiloxane, decamethylcyclopentasiloxane, dodecamethylcyclohexasiloxane,SILOL-350F—methyl silicone oil.

Table 1 shows the physical and chemical properties of the substances used in the tests.

### 2.2. Preparation of Adhesives

The initial Q2 7358 adhesive containing 57.6% solids (measured thermogravimetrically) and having a viscosity of 16.7 Pas was modified with a thermal crosslinking agent—1.5 pph dibenzoyl peroxide and toluene (to 55% solids). Then, from 0.05 to 5 pph of various silicone additives were added to the thus prepared composition (Table 2).

### 2.3. Preparation of Self-Adhesive Tapes

In order to obtain a self-adhesive tape, the adhesive had to be coated on a polyester film with a thickness of 50 g/m^2^. For this purpose, a semi-automatic PSAT coater was used. The adhesive was coated with a weight of 45 g/m^2^. The adhesive film so coated was placed in an oven and dried for 10 min at 110 °C. The conditions for thermal cross-linking and solvent evaporation are presented in Table 2.

### 2.4. Methods

#### 2.4.1. Characterization of the Pressure-Sensitive Adhesives

Potlife—dynamic viscosity of the pressure-sensitive adhesive was characterized using a DV-II Pro Extra viscometer (Brookfield, New York, NY, USA). The solid’s content was determined using a moisture analyzer (Radwag MAX 60/NP, Radom, Poland). The measurement was taken before the solvent was evaporated. The samples were heated in aluminum crucibles at 140 °C for 40 min. The basis weight of the obtained adhesive films was determined using a round punch with an area of 10 cm^2^ (Karl Schröder KG, Weinheim, Germany). The grammage test was performed for the adhesive films after evaporation of the solvent.

#### 2.4.2. Characterization of the Adhesive Tape

The peel adhesion was measured according to the AFERA 4001 standard on the Zwick-Z010 testing machine (Zwick/Roell, Germany), and the tack: AFERA 4015. The cohesion measurement of the self-adhesive tapes was carried out according to the FINAT-FTM8 standard at 20 and 70 °C.

##### Peel Adhesion Testing

Another property that determines the suitability of a pressure-sensitive adhesive is adhesion. It is very important in the technology of adhesive materials. It determines the interaction of the surfaces of the layers of phases or physical bodies, which causes the transfer of loads between them. This property is related to the action of attractive forces on the contact surface of the stuck materials [15,16].

##### Tack

This concept describes the ability of a pressure-sensitive adhesive to form a bond when briefly applied [17].

##### Cohesion Test

Cohesion is also an important property of pressure-sensitive adhesives. It is responsible for the internal strength and cohesion of the adhesive joint. All the above-mentioned self-adhesive properties are largely influenced by the following elements: type of adhesives, molecular weight of copolymer, crosslinking method (e.g., amount and type of crosslinking compound, dose, and duration of crosslinking with ultraviolet radiation), the basis weight of the adhesive layer or the temperature of the research [18,19].

##### SAFT Test (Shear Adhesive Failure Temperature)

The SAFT test was performed similarly to the cohesion test, but the temperature during the measurement was not constant, but increases from 20 to 225 °C (with a temperature increase of 1.5 °C/min. During the test, the temperature at which the sample of the tape fell off the plate was measured [20].

##### Shrinkage

The shrinkage of pressure sensitive adhesives is measured by observing the reduction in size of the adhesive film from its original size. It is related to the cross-linking process (with the type of cross-linking method and the cross-linking compound). It is an important property where shrinkage can affect the surface of the adhesive and cause it to deform. For pressure sensitive adhesives, shrinkage is defined as the change in size of the adhesive film. The result is given in percent or millimeters. The film used is PVC (polyvinyl chloride) or PET (poly (ethyl terephthalate)), onto which the adhesive film is transferred and then sticked onto a metal plate. Two cuts are made on the foil and placed at a temperature of 70 °C. Then, after a specified time, the size of the cuts made is examined. The result of the test is the arithmetic mean of eight points. In the technology of PSA, the shrinkage value above 0.5 mm or 0.5% exceeds the allowable value [21,22].

## 3. Results

The results for the pressure-sensitive adhesive without modification with silicone additives are presented in Table 3. Tack and peel adhesion values of the starting adhesive are satisfactory, as is the cohesion at room temperature and elevated temperature. Self-adhesive tapes without modification, however, show a low temperature value in the SAFT test, which reduces the range of applications of such tapes.

Figure 1 shows effect of silicone derivatives addition on the peel adhesion of silicone pressure sensitive adhesives. Black color marks PDMS OM 50 additive, red PDMS 500, green OS 300 and blue SILOL 350F. The figure also shows the result for the adhesive without modification with a dashed line (marked with the value 0). As the content of the silicone additive increases, the peel adhesion of the adhesive tape decreases. The greatest decrease was observed for the addition of SILOL 350F, where adhesion equal to 13.5 with the lowest content, while with the highest equal to 6.87 (the lowest of all). As can be seen from the figure, the addition of silicone compounds causes a decrease in the peel adhesion value relative to the initial level. Only at the lowest filling equal to 0.05 an increase in these values is observed for the additives OS 300 and SILOL 350F. This may be due to the fact that the degree of crosslinking of the adhesive with lower amounts of silicone additives is higher compared to tapes with a higher content of additives. In this case, the cross-link density of the adhesive will be lower, due to the presence of other compounds, it is not possible to form a highly cross-linked network [23,24].

The cohesion of the modified pressure-sensitive adhesives is shown in Figure 2 and Figure 3. The value of the cohesion of silicone pressure-sensitive adhesives, defined as the time to the cohesive failure, is shown in Figure 2. In general, an increase in the concentration of silicone additives increases this time. In the case of cohesion at 20 °C, the maximum value of 72 h was obtained for most of the tests. This value is identical to that of the starting sample. In the case of the maximum amount of additives (5 pph) for the addition of PDMS 500 and OS 300. It follows that the adhesive containing polydimethylsiloxane polymers with a higher viscosity (500 mm^2^/s) causes a decrease in cohesion at room temperature. In the case of cohesion at a temperature of 70 ℃, PDMS 500 additive shows the highest values compared to the other additives. This proves that the addition of polydimethylsiloxane polymers with higher viscosity (500 mm^2^/s) has a positive effect on the value of cohesion at elevated temperatures. On the other hand, the additives OS 300, SilOl 350F and PDMS OM 50 for the lowest contents in the composition (0.05–0.1 pph) show low values of cohesion. Similarly, for the highest value of 5 pph, in most cases, belt samples with additives in the range of 0.5 pph to 3 pph achieve maximum cohesion values of 72 h.

The SAFT test was performed up to a maximum temperature of 225 °C. Figure 4 shows the obtained results. Virtually every additive obtained the maximum value, but the best additive turned out to be PDMS 500 additives. As in the case of cohesion, samples containing the smallest proportion of the smallest silicone compound showed lower thermal resistance. Similarly, those with the highest value were also characterized by low resistance. In the case of the starting adhesives, this value was 190 °C. Virtually every additive increased the thermal resistance to a maximum value, but to a varying extent. For PDMS OM 50 from 0.1 pph to 3.0 pph, PDMS 500 from 0.05 pph to 3.0 pph, for OS 300: 1.0 pph, and for SILOL 350F from 1.0 pph to 3.0 pph.

The next figure (Figure 5) shows the tack results for the prepared compositions. As with peel adhesion, a small amount of OS300 and SILOL-350F additive increases the tack value. On the other hand, with an increase in the amount of silicone additive in the sample, this value decreases. Already at a value of 0.1 pph, a sudden drop in value is observed compared to the starting adhesive. Adding more silicone additives has a negative effect on the tack value, especially in the range from 0.1 to 5 pph.

Table 4 shows the shrinkage of pressure sensitive adhesives. The marking 0 indicates the original adhesive, without modification with silicone additives. The shrinkage of this adhesive is high and after just 1 h it exceeds the guidelines for adhesives and self-adhesive tapes. It is assumed that the adhesive tape should show a shrinkage of less than 0.5 mm or 0.5%. Compared to the starting adhesive, virtually any amount of silicone additive reduces shrinkage. Therefore, it has a positive effect on the product and increases the possibilities of its use.

Comparing the additives, it can be stated that the highest shrinkage is shown for tape samples containing OS 300 additive, which is a mixture of various silanes (octamethylcyclotetrasiloxane, decamethylcyclopentasiloxane, dodecamethylcyclohexasiloxane), while it is still lower than the starting adhesive, without modification. Samples modified with SILOL 750 F (methyl silicone oil) are characterized by lower shrinkage (compared to OS 300). They show an acceptable shrinkage for the contents of 3 and 5 pph. The remaining amounts in the following days exceed the threshold of 0.5%. The lowest shrinkage is characteristic for the samples modified with PDMS 50 and PDMS 500. This is not a big surprise because they are practically the same compounds (polydimethylsiloxane polymers), but they differ in viscosity. Modified PDMS samples with lower viscosities have a lower shrinkage value, which is a desirable property. These samples are characterized by a shrinkage of less than 0.5% in each range of additive concentrations. For PDMS modified samples with a higher viscosity, the shrinkage meets the tape guidelines in the range of 3 to 5 pph.

Figure 6 shows the viscosities of solvent-borne silicone adhesives containing silicone additives, determined 1 day after mixing and after 2, 3, 5 and 7 days. Tests were also performed after 30 and 90 days (Table 5). In all cases, there was an increase in viscosity.

Figure 6 shows the viscosity of pressure-sensitive adhesives depending on the silicone additive used. Figure 6a shows the viscosity (pot-life) of the additive PDMS 50, Figure 6b: PDMS 500, Figure 6c: OS 300 and Figure 6d: SILOL 350F. In all cases, a decrease in viscosity was observed compared to the starting adhesive marked in black. The most similar values (to the starting adhesive) are obtained for the lowest concentrations (0.05 and 0.1 pph). The biggest difference is observed in the case of the highest value of 5 pph. In the case of PDMS 50 and 500 additives, they show comparable values up to 30 Pas, similarly for OS 300, while the lowest values are obtained for SILOL 350F samples.

The most pronounced change in viscosity was observed at the highest concentrations. During the longer storage time: 30 and 90 days, the highest change was found for the OS 300 adhesive, which may indicate a greater reactivity of this adhesive towards the DClBPO cross-linking compound compared to the reactivity of other tested adhesives. However, the increase in viscosity of the tested silicone adhesives cross-linked with thermally reacting organic oxides, modified silicone compounds, significantly influences their application properties, enabling easy distribution after 3 months of storage at room temperature [24].

## 4. Conclusions

As a result of the research, the influence of silicone additives on the pressure-sensitive adhesive was determined. During the research, it was possible to improve the value of thermal resistance (even up to 225 °C) for all used additives (depending on the content). However, there was a decrease in the adhesive properties, i.e., the tack and the peel adhesion. Cohesion practically remained unchanged. It was also possible to reduce the shrinkage value of the adhesives to an acceptable value in the industry (less than 0.5%). Moreover, as a result of the addition of silicone compounds, the viscosity values were reduced, which is a useful property in the technology of obtaining adhesives and self-adhesive materials. As a result of the research. good quality self-adhesive tapes were obtained. The highest thermal resistance was obtained for adhesive films containing PDMS 50 and PDMS 500. Even a small concentration of this component significantly increases the thermal resistance.

## Figures and Tables

**Figure 1 materials-15-05713-f001:**
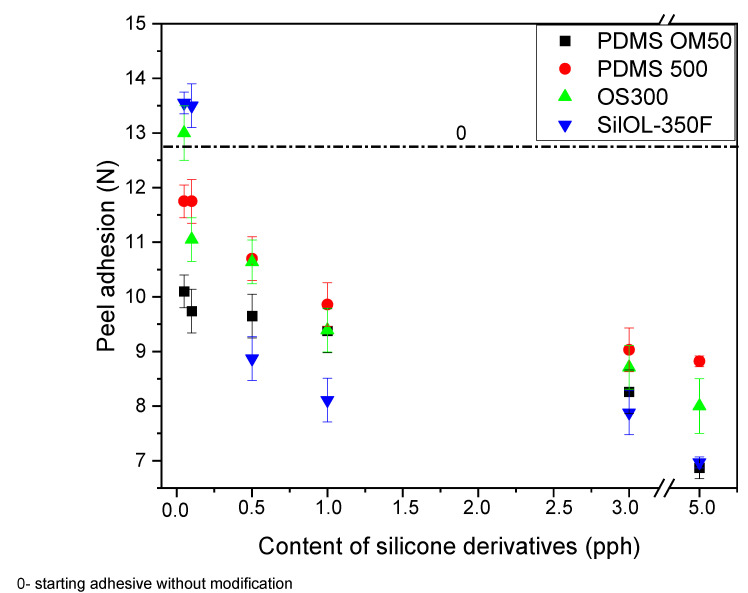
Effect of silicone derivatives addition on the peel adhesion of silicone pressure sensitive adhesives.

**Figure 2 materials-15-05713-f002:**
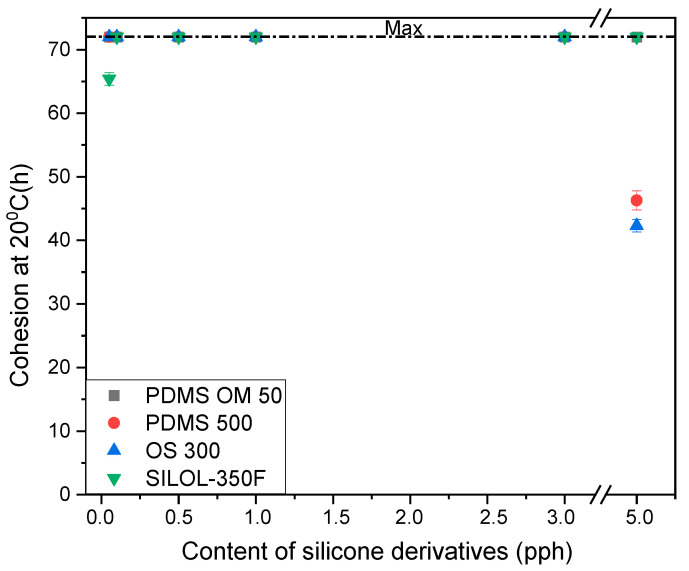
Effect of silicone derivatives addition on the cohesion at 20 °C of silicone pressure sensitive adhesives.

**Figure 3 materials-15-05713-f003:**
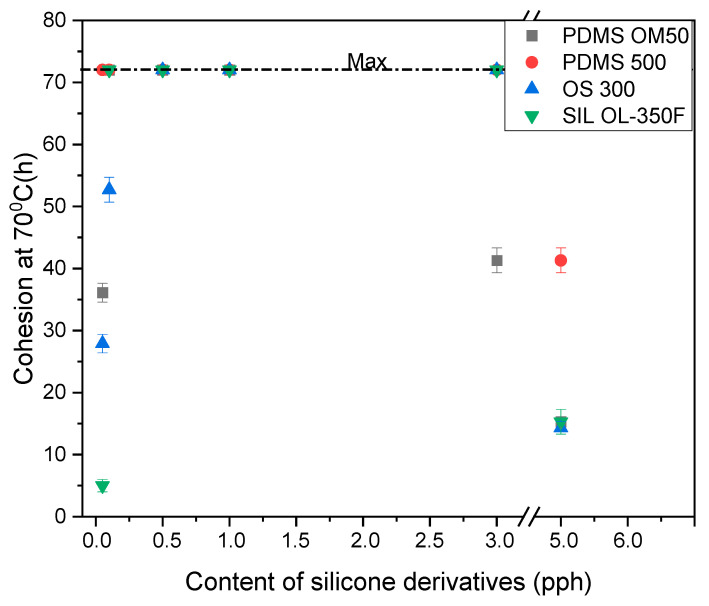
The cohesion at 70 °C of modified silicone pressure sensitive adhesives.

**Figure 4 materials-15-05713-f004:**
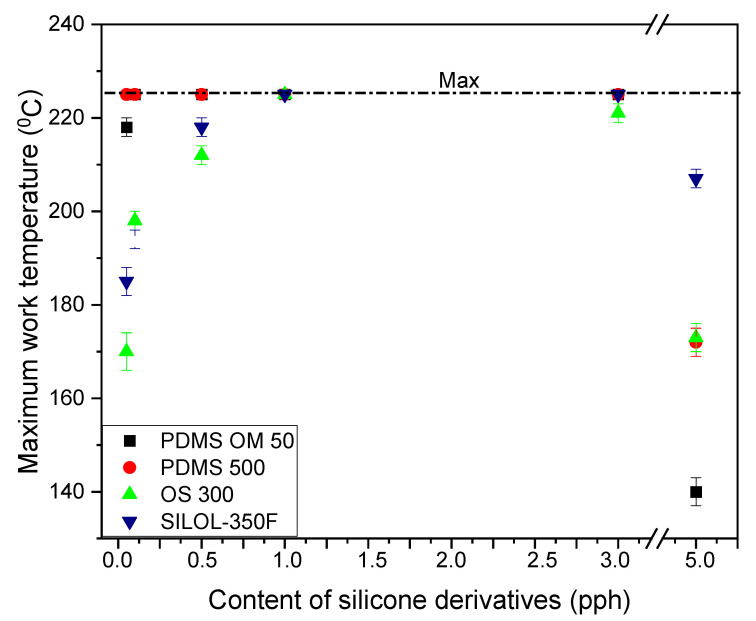
The maximum work temperature of silicone pressure sensitive adhesives with different silicone derivatives addition.

**Figure 5 materials-15-05713-f005:**
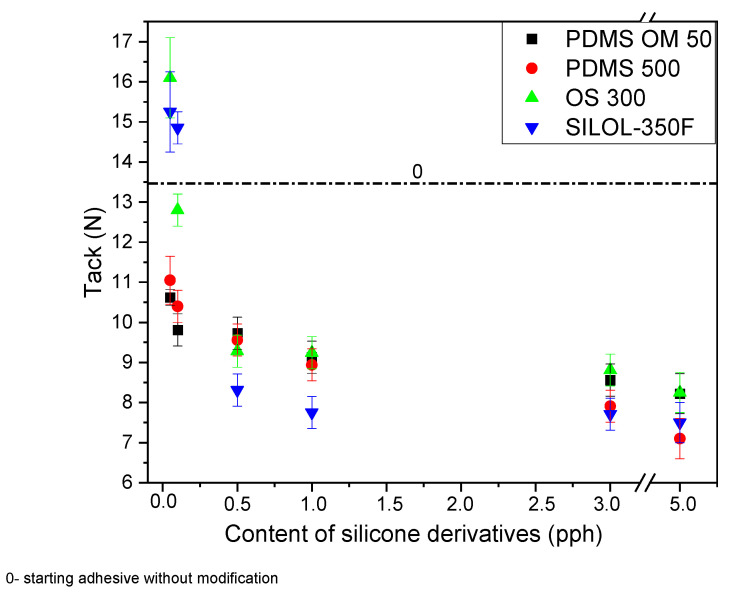
Effect of silicone derivatives addition on the tack of silicone pressure sensitive adhesives.

**Figure 6 materials-15-05713-f006:**
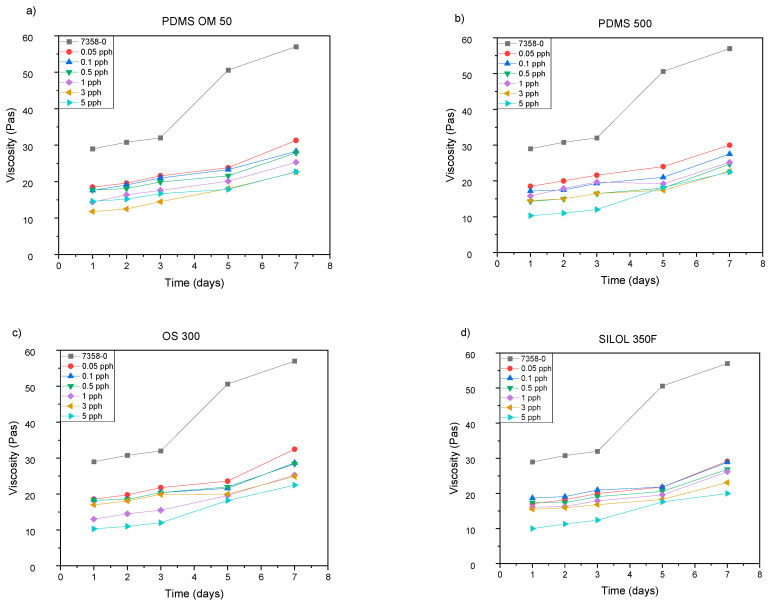
The viscosity of silicone pressure sensitive adhesives depending on the ingredient added: (**a**) PDMS OM 50, (**b**) PDMS 500, (**c**) OS 300, (**d**) SILOL350F.

**Table 1 materials-15-05713-t001:** Physical and chemical properties of substances used in the research.

Name	Appearance	Viscosity	Specific Gravity at 20 °C	Flash Point	Pour Point	Application
DOWSIL™ 7358 (Q2-7358)	Water white, clear	16.7 Pas	0.98	-	-	Pressure sensitive adhesive. Electrical insulation and bonding tapes High temperature masking tapes Plasma/flame-spray tapes Applications requiring a balance of properties emphasizing high tack.
Dichlorobenzoyl peroxide	White paste	4000 Pas	1.21 g/cm^3^	-	-	Cross-linker, polymerization initiator.
Toluene	Colorless liquid	0.56 mPa s	0.87 g/cm^3^	480 °C		Organic solvent.
PDMS OM 50	Clear colorless liquid	50 mm^2^/s	0.96 g/cm^3^	>300 °C	−54 °C	They are used in applications that require fluid properties in harsh environments. In mechanically demanding applications they provide excellent shear stability and good lubrication or slip.
PDMS 500		500 mm^2^/s	0.97 g/cm^3^	>302 °C	−47 °C	
OS 300	Colorless liquid	-	No data approx 0.970 g/cm^3^	>120 °C	-	Universal (oil) silicone grease that eliminates friction and material resistance. The agent is used in many industries, wherever there are problems with adhesion or high friction. It is resistant to the influence of external factors.
SILOL 350F	Liquid, colorless,	-	-	-	-	Used in medicine to treat burns, it protects the skin against external factors.

- = no data.

**Table 2 materials-15-05713-t002:** The conditions of preparation of adhesives and adhesive films.

Name of Additives	Content of Additives (pph)	ES ^1^: Time (min)/Temperature (°C)	Coat Weight ^2^ (g/m^2^)
PDMS OM 50	0.05	10/110	45
	0.1		
	0.5		
	1		
	3		
	5		
PDMS 500	0.05	10/110	45
	0.1		
	0.5		
	1		
	3		
	5		
OS 300	0.05	10/110	45
	0.1		
	0.5		
	1		
	3		
	5		
SILOL-350F	0.05	10/110	45
	0.1		
	0.5		
	1		
	3		
	5		

^1^ ES, evaporation of solvent; ^2^ coat weight of adhesive film.

**Table 3 materials-15-05713-t003:** Q2 7358 pressure-sensitive adhesive results without modification.

Peel Adhesion [N/25 mm]	Cohesion [h]		SAFT (°C)	Tack [N]
	20 °C	70 °C		
12.75	>72	>72	192	10.45

**Table 4 materials-15-05713-t004:** Shrinkage of pressure-sensitive adhesives: marked with 0—adhesive without modification, (**a**) PDMS OS 50, (**b**) PDMS 500, (**c**) OS 300, (**d**) SILOL 350F.

(a)												
Shrinkage (%)												
Content of Additives (pph)	10 min	30 min	1 h	3 h	8 h	24 h	2 Days	3 Days	4 Days	5 Days	6 Days	7 Days
**0 ***	**0.42**	**0.41**	**0.64**	**0.90**	**0.96**	**1.02**	**1.10**	**1.14**	**1.33**	**1.33**	**1.33**	**1.33**
0.05	0.11	0.32	0.35	0.40	0.42	0.44	0.48	0.48	0.48	0.48	0.48	0.48
0.1	0.11	0.22	0.25	0.30	0.32	0.33	0.41	0.44	0.45	0.45	0.45	0.45
0.5	0.16	0.25	0.25	0.29	0.31	0.33	0.37	0.39	0.40	0.40	0.40	0.40
1	0.14	0.22	0.29	0.32	0.37	0.38	0.39	0.40	0.40	0.40	0.40	0.40
3	0.11	0.11	0.13	0.17	0.20	0.26	0.28	0.30	0.32	0.35	0.35	0.35
5	0.03	0.04	0.04	0.05	0.05	0.05	0.06	0.06	0.07	0.07	0.08	0.08
**(b)**												
**Shrinkage (%)**												
**Content of Additives (pph)**	**10 min**	**30 min**	**1 h**	**3 h**	**8 h**	**24 h**	**2 Days**	**3 Days**	**4 Days**	**5 Days**	**6 Days**	**7 Days**
0.05	0.28	0.30	0.36	0.39	0.42	0.48	0.50	0.59	0.60	0.69	0.69	0.69
0.1	0.21	0.23	0.24	0.27	0.28	0.42	0.51	0.56	0.6	0.67	0.69	0.69
0.5	0.23	0.24	0.25	0.39	0.31	0.34	0.46	0.51	0.58	0.61	0.67	0.67
1	0.21	0.24	0.26	0.30	0.38	0.40	0.49	0.54	0.60	0.63	0.63	0.63
3	0.06	0.07	0.07	0.08	0.09	0.09	0.09	0.10	0.10	0.12	0.15	0.15
5	0.03	0.04	0.04	0.05	0.05	0.06	0.06	0.06	0.07	0.07	0.08	0.09
**(c)**												
**Shrinkage (%)**												
**Content of additives (pph)**	**10 min**	**30 min**	**1 h**	**3 h**	**8 h**	**24 h**	**2 Days**	**3 Days**	**4 Days**	**5 Days**	**6 Days**	**7 Days**
0.05	0.31	0.36	0.40	0.48	0.50	0.53	0.66	0.70	0.74	0.87	0.89	0.89
0.1	0.21	0.26	0.27	0.32	0.38	0.41	0.45	0.52	0.60	0.64	0.64	0.64
0.5	0.21	0.25	0.30	0.31	0.37	0.40	0.42	0.46	0.48	0.51	0.55	0.58
1	0.31	0.31	0.32	0.35	0.39	0.39	0.40	0.40	0.41	0.48	0.51	0.51
3	0.20	0.23	0.24	0.27	0.27	0.30	0.32	0.34	0.37	0.41	0.46	0.46
5	0.1	0.11	0.13	0.15	0.19	0.21	0.25	0.30	0.30	0.33	0.34	0.34
**(d)**												
**Shrinkage (%)**												
**Content of Additives (pph)**	**10 min**	**30 min**	**1 h**	**3 h**	**8 h**	**24 h**	**2 Days**	**3 Days**	**4 Days**	**5 Days**	**6 Days**	**7 Days**
0.05	0.42	0.48	0.50	0.58	0.63	0.65	0.72	0.76	0.81	0.87	0.87	0.87
0.1	0.39	0.40	0.46	0.50	0.54	0.60	0.63	0.68	0.70	0.74	0.74	0.74
0.5	0.29	0.30	0.31	0.36	0.40	0.43	0.47	0.50	0.57	0.62	0.65	0.65
1	0.28	0.29	0.30	0.30	0.31	0.32	0.36	0.38	0.40	0.41	0.41	0.41
3	0.10	0.13	0.15	0.25	0.27	0.29	0.32	0.36	0.38	0.38	0.39	0.39
5	0.06	0.06	0.06	0.06	0.07	0.07	0.08	0.08	0.09	0.09	0.10	0.12

* 0—(adhesive without modification).

**Table 5 materials-15-05713-t005:** Viscosity of pressure-sensitive adhesives after 30 and 90 days.

	Content of Additives (pph)	Ƞ30, (Pas)	Ƞ90, (Pas)
PDMS OM 50	0.05	33.0	38.0
	0.1	32.4	36.5
	0.5	32.0	35.8
	1	31.4	34.3
	3	31.1	34.0
	5	30.4	33.8
PDMS 500	0.05	34.0	39.2
	0.1	33.8	38.6
	0.5	33.2	37.6
	1	32.4	37.0
	3	31.4	36.1
	5	31.0	35.4
OS 300	0.05	36.0	41.0
	0.1	34.5	40.0
	0.5	34.0	39.8
	1	33.2	37.2
	3	32.0	35.8
	5	29.0	35.0
SILOL 750F	0.05	31.0	35.4
	0.1	30.5	34.7
	0.5	29.4	33.9
	1	27.4	32.4
	3	26.7	31.6
	5	25.4	30.4

## Data Availability

Not applicable.

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
