# Peer review of "Influence of Silicone Additives on the Properties of Pressure-Sensitive Adhesives"

_materials, 2022, doi:10.3390/ma15165713_

Round 1
Reviewer 1 Report
I think the work is interesting and useful. The manuscript can be accepted after minor reversion.
1 Please make clear the purpose and importance of adding the silicone additives.
2 Please make clear the innovation of this paper.
3 Some capital letter abbreviations such as SAFT test should give the full names when they first apper.
Author Response
I think the work is interesting and useful. The manuscript can be accepted after minor reversion.
1 Please make clear the purpose and importance of adding the silicone additives.
We agree with the Reviewer's opinion. The purpose has been supplemented in the text at the end of the introduction.
2 Please make clear the innovation of this paper.
The innovation of the article is the determination of the influence of various silicone additives: polydimethylsiloxanes polymers and various silicone oils. During the literature review, no other documents were found that would describe this issue. We added this information to the article.
3 Some capital letter abbreviations such as SAFT test should give the full names when they first apper.
We agree with the Reviewer's opinion. The abbreviation was expanded in the text.
Finally, we hope that corrections made in the manuscript fulfill reviewer suggestions and allow editor to make positive decision about acceptation of our contribution for publishing in this journal.
With regards,
Karolina Mozelewska
Adrian Krzysztof Antosik
Reviewer 2 Report
Colloidal materials have always been one of the indispensable materials in people's production and life. I personally do not have much research on such materials, so I may not be able to judge this manuscript very accurately. However, based on my interest in materials and research accumulation, I give some suggestions for improvement of this manuscript from my personal perspective: the manuscript seems to lack the characterization of materials, For example, from the perspective of molecular dynamics, explain how silicone additives affect the properties of materials at the material structure level. In conclusion, I think this manuscript is acceptable.
Author Response
The authors would like to thank to the Reviewer and truly appreciate his comments, questions and corrections. Please find the detailed answers below.
Colloidal materials have always been one of the indispensable materials in people's production and life. I personally do not have much research on such materials, so I may not be able to judge this manuscript very accurately. However, based on my interest in materials and research accumulation, I give some suggestions for improvement of this manuscript from my personal perspective: the manuscript seems to lack the characterization of materials, For example, from the perspective of molecular dynamics, explain how silicone additives affect the properties of materials at the material structure level. In conclusion, I think this manuscript is acceptable.
We want to thank the reviewer for the suggestion, we took them into account as best we could. We hope we did it right.
Finally, we hope that corrections made in the manuscript fulfill reviewer suggestions and allow editor to make positive decision about acceptation of our contribution for publishing in this journal.
With regards,
Karolina Mozelewska
Adrian Krzysztof Antosik
Reviewer 3 Report
The paper seeks to introduce an approach ‘’ Influence of silicone additives on the properties of pressure-sensitive adhesives”. However, the authors should consider improving upon the quality to further highlight and emphasize.
1. Based on the understanding of what should be included in the abstract, consider adding one or two lines introducing the problem this article seeks to address.
2. Also, add one or two lines at the end of the abstract highlighting the significance of the study.
3. Add actual figures of your findings to the abstract to give a proper summary of what was achieved.
4. The introduction needs to be improved by relating to the mechanics of the studied materials and their mechanical characteristics. The references to be included are: 10.3390/polym14132662, 10.1016/j.jiec.2022.06.023, 10.1016/j.porgcoat.2022.107015.
5. Tabulate all the materials used under the materials section with their physical and chemical properties.
6. The space between 50 and g/m2 is too wide. Leave just a space between the variable and its respective unit.
7. The name ascribed to the figure in the description should commensurate with the name in the caption text. The author referred to figure 1 as the first figure which is not right in line 180. Make sure all figures and tables are uniformly named both in caption text and in the description or in the discussion.
8. Not a single diagram about the various characterization. Try and include diagrams to appreciate how the test was done.
9. One standard of writing style should be adopted. The author used Figure and Fig. at the same time in line 197 and 199 which is not acceptable. Consider using one style
10.Some of the variables and their corresponding units are put together. Put space between each.
Author Response
The authors would like to thank to the Reviewer and truly appreciate his comments, questions and corrections. Please find the detailed answers below.
- Based on the understanding of what should be included in the abstract, consider adding one or two lines introducing the problem this article seeks to address.
We agree with the Reviewer's opinion. A few sentences have been added to the text about the problem.
- Also, add one or two lines at the end of the abstract highlighting the significance of the study.
We agree with the Reviewer's opinion. A sentence has been added at the end of the abstract.
- Add actual figures of your findings to the abstract to give a proper summary of what was achieved.
We agree with the Reviewer's opinion. Numbers added to the abstract. We hope it is right.
- The introduction needs to be improved by relating to the mechanics of the studied materials and their mechanical characteristics. The references to be included are: 10.3390/polym14132662, 10.1016/j.jiec.2022.06.023, 10.1016/j.porgcoat.2022.107015.
In the presented work, no mechanical tests were performed, fatigue strength and tensile tests were not assessed (as in the case of 10.3390/polym14132662). Also, the particle size and zeta potential, kinetic models were not determined (10.1016/j.jiec.2022.06.023). It is therefore difficult to relate to them in our work.
- Tabulate all the materials used under the materials section with their physical and chemical properties.
We agree with the Reviewer. A table with properties has been added to the manuscript.
- The space between 50 and g/m2 is too wide. Leave just a space between the variable and its respective unit.
We have made corrections. We hope it is right.
- The name ascribed to the figure in the description should commensurate with the name in the caption text. The author referred to figure 1 as the first figure which is not right in line 180. Make sure all figures and tables are uniformly named both in caption text and in the description or in the discussion.
We have made corrections. We hope it is right.
- Not a single diagram about the various characterization. Try and include diagrams to appreciate how the test was done.
The characteristics and descriptions of some figures and tables have been changed.
- One standard of writing style should be adopted. The author used Figure and Fig. at the same time in line 197 and 199 which is not acceptable. Consider using one style
We have made corrections.
10.Some of the variables and their corresponding units are put together. Put space between each.
Thank you very much for this attention. We put the gaps between the data and the units.
Finally, we hope that corrections made in the manuscript fulfill reviewer suggestions and allow editor to make positive decision about acceptation of our contribution for publishing in this journal.
With regards
Karolina Mozelewska
Adrian Krzysztof Antosik